# Biosensing Using SERS Active Gold Nanostructures

**DOI:** 10.3390/nano11102679

**Published:** 2021-10-12

**Authors:** Gour Mohan Das, Stefano Managò, Maria Mangini, Anna Chiara De Luca

**Affiliations:** Laboratory of Biophotonics and Advanced Microscopy, Second Unit, Institute of Experimental Endocrinology and Oncology “G. Salvatore” (IEOS), National Research Council (CNR), Via P. Castellino 111, 80131 Naples, Italy; gourmohan.das@ieos.cnr.it (G.M.D.); maria.mangini@ieos.cnr.it (M.M.)

**Keywords:** SERS, biosensing, gold nanoparticles

## Abstract

Surface-enhanced Raman spectroscopy (SERS) has become a powerful tool for biosensing applications owing to its fingerprint recognition, high sensitivity, multiplex detection, and biocompatibility. This review provides an overview of the most significant aspects of SERS for biomedical and biosensing applications. We first introduced the mechanisms at the basis of the SERS amplifications: electromagnetic and chemical enhancement. We then illustrated several types of substrates and fabrication methods, with a focus on gold-based nanostructures. We further analyzed the relevant factors for the characterization of the SERS sensor performances, including sensitivity, reproducibility, stability, sensor configuration (direct or indirect), and nanotoxicity. Finally, a representative selection of applications in the biomedical field is provided.

## 1. Introduction

Surface enhanced Raman scattering (SERS) can be defined as amplified Raman scattering by the presence of plasmonic nanostructures (generally metallic nanoparticles) in the proximity of the analyte molecules [1]. Upon excitation with appropriate light interacting with the sub-nanometer metallic structures, the collective oscillation of conduction electrons is observed, generating an ultra-strong electromagnetic (EM) near-field in the proximity of the nanostructure surface. This phenomenon is known as localized surface plasmon resonances (LSPRs) and it is at the foundation of SERS spectroscopy [2]. SERS spectroscopy has particular advantages for biosensing applications [3]: (i) it allows reconstruction of a detailed spectral profile providing all the structural, compositional, and conformational features of the analyte molecules; (ii) it enables multiplex detection with single wavelength excitation, therefore several molecules can be monitored at the same time; (iii) it is not destructive; (iv) it allows measurements in biological fluids since the water spectrum is rather weak; (v) SERS is resistant to photo-bleaching and photo-degradation compared with fluorescence and is suitable for long-term monitoring; and (vi) SERS does not require high sample concentrations. In other words, SERS spectroscopy combines the intrinsic advantages of Raman spectroscopy with high sensitivity. Indeed, the million-fold enhancement enables SERS detection down to a single molecule level [4,5].

The key element of SERS spectroscopy is the realization of the metallic nano-substrate. The development of several bottom-up chemical synthesis procedures or advanced top-down nanofabrication technologies allows for obtaining a variety of nanostructures with tunable geometries [5]. Indeed, SERS-active nanostructures can be designed with different sizes, shapes, compositions, aggregations, and coatings, allowing the possibility of response tuning for specific detection purposes. The SERS substrate can be optimized for the near-infrared excitation laser, thus avoiding auto-fluorescence from biological samples and minimizing their photo damage. Within the group of noble metal nanostructures, gold-based nanostructures or gold nanoparticles (AuNPs) are mostly used for biosensing applications due to their biocompatibility, tunability, their unique optical and electronic properties, together with their nontoxic nature when compared to other nanomaterials, such as metal oxides or carbon-based nanomaterials in general [6,7,8]. As a result, there have been several impressive developments and applications of SERS active gold nanostructures for biosensing applications [9,10,11].

In this review, we discuss in detail the analytical and sensing capabilities of SERS in biomedicine. First, a very fundamental and brief introduction of Raman scattering, followed by the origin of SERS is provided. Then, a discussion on the key elements of the SERS sensors, including fabrication/synthesis, sensitivity, reproducibility, detection configurations, and toxicity of the SERS-active substrates, with a focus on gold nanostructures, is reported. Finally, different biomedical and biosensing applications of SERS are highlighted.

## 2. Raman and Surface Enhanced Raman Scattering (SERS) Techniques

Raman spectroscopy is an optical vibrational spectroscopic technique based on the inelastic scattering of light [12]. The energy lost in the scattering events by the photons is called the Raman shift and is represented by wavenumbers. [13]. The Raman cross-section can determine the efficiency of a particular molecule, with a random orientation with respect to incident field polarization, to scatter light. If the rate of change of polarizability with the vibration is not zero, then the molecule is Raman active [13]. Raman is an intrinsically very weak phenomenon, approximately one photon in one million is Raman scattered [14]. However, the Raman scattering generated by analyte molecules can be strongly enhanced by nearby nanostructured metallic surfaces. This phenomenon is called SERS, rather than simple Raman, to emphasize the amplification effect due to the presence of the plasmon active substrate [13].

In 1974, Fleischmann et al., first reported the SERS phenomenon describing an unexpectedly higher Raman signal of the pyridine monolayer adsorbed on a roughened silver electrode [15]. Later on, Van Duyne and Moskovits clarified the origin and the mechanisms at the basis of the SERS enhancement [16,17,18]. According to the literature, the SERS signal enhancement originates from two mechanisms: electromagnetic enhancement and chemical enhancement [19].

When a molecule is placed on or very close (within 2–3 nm) to the surface of a metal nanostructure and illuminated with appropriate laser light, this supports the excitation of surface plasmons on the substrate. LSPRs, thanks to the collective oscillation of conduction electrons, create strong electromagnetic near-field intensities on the substrate surface. Then, any molecule adsorbed on the substrate surface, due to the huge local field intensity, generates a higher number of stokes photons (EM enhancement). Figure 1a,b shows the schematic of LSPR from plasmonic nanoparticles and distance-dependent SERS intensity. In the presence of metallic nanoparticles, the localized surface plasmon (LSP) generates high-intensity local hotspots on the surface of the plasmonic nanoparticles when interacting with incident laser light. The comparison between normal Raman scattering and two separate contributions to EM enhancement are shown in Figure 1c,d, respectively; (i) Local field (or near field) enhancement: excitation of surface plasmons generates a powerful spatial localization which amplifies the laser light in ‘nanometer’ regions, known as hot-spot; (ii) Re-radiation enhancement: the existence of the metallic structure near the molecule also changes the efficiency with which the molecule radiates Raman power, as the power radiated by a dipole relies on the surroundings in which it is placed.

Therefore, the expected power of the Stokes signal of molecules adsorbed on a metallic nanoparticle, due to the EM enhancement is given by [12]
(1)$$P^{SERS}(\nu_{s})=N^{\prime }\sigma_{}^{R}I(\nu_{L}){\left|A(\nu_{L})\right|}^{2}{\left|A(\nu_{S})\right|}^{2}$$
where $\sigma^{R}$ describes increased Raman scattering cross-section of a molecule adsorbed on the nanosphere, and $N^{\prime }$ is the number of molecules that are involved in the SERS process. $A(\nu_{L})$ and $A(\nu_{s})$, respectively, represent the incident laser and Stokes radiation field enhancement factors due to nanoparticle. The enhancement factor (*EF*) of the Raman scattering signal in the SERS technique as compared with the conventional technique can be approximated as
(2)$$EF\approx \eta {\left|A(\nu_{L})\right|}^{2}{\left|A(\nu_{s})\right|}^{2}$$

Here, η is the factor that depends on the geometry of the nanostructure. From this equation, it can be seen that the enhancement factor scales as the square of the local electric field intensity at any of the hot spots of the metallic nanoparticle, and this factor is particularly strong when excitation and scattered light frequencies are in match with the frequency of the LSP oscillation frequency (at resonance condition). In addition, the enhancement factor decreases with increasing distance from the nanoparticle due to the rapid decay of local electric field intensity.

The chemical enhancement is strongly dependent on the type of Raman probe molecule as it is due to the physicochemical interaction between molecules with the SERS substrate. This mechanism is always considered a short-range effect because it is only possible when the molecule and SERS substrate are in contact or at a distance of few angstroms. The chemical enhancement contributes much less than the EM enhancement, in the range of 10^2^–10^4^, depending on different chemical structures, specific vibration, and interaction with a metallic surface [20,21,22,23]. During the process of charge transfer between substrate and molecule, at first, an electron in the valance band of the substrate is stimulated by the incident light. As a result, a temporal electron in the conduction band (CB) and a hole in the valence band (VB) are generated. After that, the excited electron is quickly transferred from the CB of the substrate to the matching energy level above the LUMO of the molecule through the resonant tunneling. The electron then will transit back to the substrate and recombine to the hole of the VB (illustrated in Figure 1e). During this process, a vibrational quantum of energy will transfer to the vibrational level of the molecule and a Raman photon will be radiated from the molecule at that time at some vibrational state of the molecule. Now, the intensity of the Raman transition is given by
(3)$$I_{SERS}=[8\pi {\left(\omega_{0}\pm \omega_{IF}\right)}^{4}I_{0}/9c^{4}]\sum \alpha^{2}$$
where *α* is the polarizability tensor (nine components) given by *α* = *A* + *B* + *C*. Here, *A* is coming from molecular polarizability tensor, *B* is the charge transfer between the molecule to the substrate, and *C* is the charge transfer between the substrate to the molecule. *I_0_* is the incident laser intensity at $\omega_{0}$. $\omega_{IF}$ is a molecular transition frequency between the two states.

## 3. SERS Substrates Used for Biosensing

At the core of SERS biosensing is the interaction between the analyte molecule and the selected substrate. Therefore, the selection/fabrication of the SERS substrate is a key element to determine the performance of the sensor. The first SERS substrate was fabricated using the electrochemically roughened metal electrodes, and the main limitations are the reduced control on plasmon resonance as well as the hotspot due to the aggregation with low tenability, even though one can produce a large-area SERS substrate using this technique [15]. Due to the development of nanotechnologies, the use of metal nanoparticles as a highly sensitive and more controlled SERS substrate for biosensing has increased exponentially. Gold and silver are the most commonly used materials for the fabrication of SERS active substrates for enhancing the Raman scattering signal in the visible region. The fundamental mode of the LSPR band in the visible region depends upon the dielectric constant of gold and silver. Even though silver has the largest plasmon resonance in an easily accessible spectral region, due to its strong toxicity to living systems, Ag is seldom used for in vivo analysis, but it is good for in vitro ultrasensitive detection [24,25]. Whereas the Au provides more chemical stability, Au nanostructures are good for excitation in visible and near-infrared regions and are commonly used in intracellular or in vivo studies due to their excellent biocompatibility [26,27].

The most commonly used nanoparticle synthesis techniques are the wet chemistry method and the laser ablation technique [28,29]. Spherical metallic colloidal nanoparticles generally used in SERS experiments can be synthesized by reduction of metal salts using the wet chemistry method [30,31] (see Figure 2a). Nanoparticles can be alternatively synthesized using laser ablation [29]. A metal target is placed at the bottom of a solution and a pulsed laser is nearly focused at its surface; the heating and photoionization processes cause the metal to change the state of aggregation, forming liquid drops, vapors, or a plasma plume. Even though this method is less easy than the previous one, using this technique one can still fabricate the plasmon active substrate without capping agents, which allows an easier functionalization of nanoparticles [19].

The size and shape of the nanoparticles strongly influence the performances of the substrate for sensing applications. If the size of the metal nanoparticles on the SERS substrate is too large then the multipole (non-radiative modes) excitation occurs rather than a dipole. Due to this, the overall enhancement in the Raman scattering will reduce. Based on the literature, nanoparticles that have a size between 10–100 nm are more suitable for SERS experiments [34,35]. The shape of the nanoparticle is an additional key parameter to consider for improving the SERS performances in metal nanoparticle suspensions. Besides nanospheres, many other shapes, including nanorods, nanoprisms, nanocubes, nanoplates, nanostars, etc. have been proposed and demonstrated for different biosensing applications [36,37]. Additionally, core-shell nanoparticles, alloy nanoparticles, core-satellite nanoparticles, and yolk-shell and core molecule-shell have been utilized in SERS investigations to improve sensor performance [38,39,40,41]. Some popular shapes of metal nanostructures that are used very often for different biosensing experiments have been shown in Table 1. The shape and size of colloidal metal nanoparticles always have a strong influence on the EF of the SERS signal [1,42]. The obtained EF is summarized in Table 1, but a fair comparison between different shapes is complicated by the use of different analyte molecules.

In metal nanoparticle suspensions for SERS sensing applications, it is important to control the nanoparticle aggregation and distribution. The colloidal nanoparticle solution can be directly spotted on a glass slide followed by evaporation of the solution (drop-casting) [61,62] (Figure 2a). This method is easy and fast but without specific control of the nanoparticle distributions and signal reproducibility. However, to obtain an ordered nanoparticle distribution over a slide, typically a glass surface, the silanization protocol with (3-Aminopropyl) trimethoxysilane (APTMS) can be applied [63] (Figure 2a). In this case, a strong interaction between –NH_2_ groups and the nanoparticles occurs which helps to prevent spontaneous coalescence (it is observed in simple drop-casting on the glass substrate) and this protocol is valid for any type of dielectric substrate like glass, ITO, silicon, etc. [64,65,66].

Alternative strategies for in-liquid SERS detection are currently based on chemically-driven aggregation or optical trapping of metal nanoparticles in the presence of the target molecules [67]. An interesting example is the use of contactless manipulation methods, like laser tweezers, where the optical forces are used. A nanoparticle is subjected to two forces: (i) A gradient force that is attractive towards the high-intensity region of the laser beam if the excitation wavelength is longer than the surface plasmon resonance of the nanoparticle and repulsive in the opposite case; (ii) a radiation pressure force that propels the nanoparticle along the propagation direction of the beam [68,69].

In the last few years, researchers have developed different types of metallic nanostructures as SERS-active substrates via advanced nanofabrication techniques [32,33,70,71,72,73,74,75,76]. Electron beam lithography is one of the most popular techniques used for the fabrication of the SERS substrate (shown in Figure 2b), where electron beam of 10–30 keV incident on the silicon wafer (covered by either positive and negative resist). After the etching using an electron beam, there are two ways to fabricate the SERS substrate. One process consists of the chemical etching that follows the electron beam exposing, the dissolution of the remaining resist layer, and deposition of metal, and finally, the substrate is covered with metal. Another technique involves metal deposition immediately after the electron beam exposition. After removal of the photo-resist layer, the substrate will present a series of isolated nanoparticles, separated by regions where only bare Si substrate is exposed [32]. The main advantage of this process is the possibility to control the size, shape, and inter-particle distance between the nanoparticles. In addition, with the electron beam lithography, optical lithography has been also used for the fabrication of SERS active substrates [70]. Different SERS active substrates fabricated by lithography techniques are shown in Figure 3. Generally, all these techniques provide several advantages in terms of substrate reproducibility and sensitivity but require serial on-surface fabrication steps that are error-prone and time-consuming. Furthermore, costly instrumentation and procedures, including the use of a cleanroom facility, are required and it is very difficult to cover the large area of SERS devices using these approaches.

Recently, it has been shown that processes based on self-assembly represent an easier, quicker, and cheaper bottom-up option than nanolithography for the fabrication of regular micro-and nanopatterns [74,75,76,77] (see Figure 2c). Indeed, several SERS sensing devices have been fabricated using the nanosphere lithography technique. Cusano et al., demonstrated that polystyrene spheres (with diameters ranging from 200 to 1000 nm) self-assembled at the air-water interface into ordered hexagonal close-packed arrays and covered by a gold conformal layer can work as an efficient SERS substrate (basic close-packed array, CPA, monolayer) [75]. Their proposed fabrication procedure was inexpensive, simple to implement, and well-ordered. The substrates can be deposited on a large area, on any type of surface including the fiber tip, and can be efficiently exploited to create optical fiber SERS probes [76].

## 4. Performances of SERS Based Biosensors

The basic performance of any biosensor can be explained in terms of sensitivity (limit of detection), reproducibility, and selectivity (chemical specificity). Generally, the sensitivity of a SERS-based sensor is determined by measuring the *EF* for the selected SERS substrate, and it depends on several experimental parameters [78,79], (i). characteristics of laser excitation, (ii). detection setup, in particular: scattering configuration (e.g., backscattering geometry), (iii). SERS substrate characteristics, in particular: material, geometry, orientation with respect to the incident beam direction, and polarization, (iv). Intrinsic properties of the analyte, (v). SERS analyte adsorption properties, in particular: adsorption efficiency (surface coverage), distance from the surface, adsorption orientation (fixed or fluctuating), and modification of Raman polarizability induced by adsorption.

The determination of the magnitude of EF is not very straightforward, a detailed comprehensive study was reported by Le Ru et al. [78,79]. From the SERS substrate point of view, the EF can be defined as
$$EF=\frac{I_{SERS}/N_{surf}}{I_{RS}/N_{vol}}$$
where $I_{SERS}$ and $I_{RS}$ are the SERS and normal Raman scattering intensity for the same probe molecule. $N_{vol}(=c_{RS}.V)$ is the average number of molecules of the solutions with a concentration $c_{RS}$ in the scattering volume (*V*) for the Raman (non-SERS) measurements. Here, the value of *V* can be estimated by *V* = *Ah*, where *A* is the area of illumination or laser-spot size and *h* represents the thickness of an ultrathin film or solution layer near the ideal focal plane (*z* = 0, where the Raman intensity is the highest) [80]. $N_{surf}$ is the average number of adsorbed molecules in the scattering volume for the SERS experiments and can be calculated using the expression, $N_{surf}=RA_{g}/\sigma$, where *A_g_* is the known geometric surface area in the illuminated spot and *σ* is the surface area occupied by the molecule assuming full monolayer adsorption. Meanwhile, according to the literature, it is highly recommended to analyze multiple spots of the SERS substrate, and the average value should be preferable to report for a more consistent and accurate *EF* value.

To provide an estimate of the advantage of SERS sensing with respect to normal Raman sensing, the SERS gain, *G*, can be additionally measured [80]. Indeed, the SERS gain, *G*, is calculated as the ratio between the SERS (*I_SERS_*) and Raman (*I_Raman_*) intensity of a reference band, normalized to the different powers (*P_SERS_*, *P_Raman_*), integration times (*t_SERS_*, *t_Raman_*), and molecular concentrations (*C_SERS_*, *C_Raman_*) used in the experiment: $G=\frac{I_{SERS}/(P_{SERS}\cdot t_{SERS}\cdot C_{SERS})}{I_{Raman}/(P_{Raman}\cdot t_{Raman}\cdot C_{Raman})}$. *G* provides quantitative information on the signal gain that one has to expect from a specific SERS sensor with respect to a reference Raman experiment and it is free from any overestimation error made when evaluating the probed molecules as in the case of the EF. Indeed, *G* depends on the aggregate size and not on the target molecule concentration. When the nanoparticle aggregate has filled the scattering volume probed by the laser beam, *G* saturates, since molecules outside such a volume provide a negligible contribution to the total optical signal [67].

The term SERS reproducibility means the SERS signal from a particular molecule should be comparable within a certain error range in almost similar signal record conditions. However, it is observed that the SERS has been criticized for its poor reproducibility in different practical applications. Many important factors play a major role to determine the reproducibility, like instrumental conditions, sample preparations, high heterogeneity of the EM enhancement, as well as distribution of the sample over the substrate [80]. The spectral intensity could be varied due to changes in sample position, laser power, and drift in optical alignment. To overcome this problem, it is very important to routinely calibrate the Raman instrument during the signal measurement. Sometimes the spectral reproducibility can be maintained using a lower numerical aperture microscope objective lens that helps to average a large number of molecules in the excitation volume [81,82]. Another important issue is the proper selection of materials for SERS substrate fabrication. Indeed, oxidation of the substrate material can directly influence the LSPR properties as well as the SERS spectral intensity. This is particularly true for silver-based SERS substrates in the air [83]. Therefore, chemically robust and stable materials are useful for a reproducible SERS spectrum [84,85]. A critical role is played by the uniformity of the SERS substrate to ensure the reproducibility of the SERS signal. In this matter, the SERS substrate fabricated with well-controlled nanostructures is always better than the randomly oriented nanostructures [32,33,70,71,72,73,74,75,76]. In this matter, another crucial concern is the structural instability of the hotspot in the SERS substrate, especially for the colloidal nanostructures. These structural changes originate from the melting and diffusion of surface atoms due to the laser illumination on the SERS substrate. As a result, the change in nanogap, size, and shape of the nanoparticles is observed in the case of nanostructures with star, cube, and rod shapes. Even though these nanostructures are well known for high enhancement factors, they are also very prone to surface diffusion to decrease their surface energy by reshaping the nanoparticles into a sphere-like stable structure. To avoid this problem, one can use the core-shell nanostructure, and also decreasing laser power density and exposure time can be useful for maintaining the reproducibility of the SERS substrate [86]. Sometimes, the SERS community uses microfluidic devices to obtain an alternative strategy for improving SERS reproducibility from a commercial point of view [87].

The field enhancement distribution on a SERS substrate is highly inhomogeneous and mainly localized in the hot spots. From a structural point of view, the hot spots are often identified as nanogaps between nanoparticles, therefore the dimensions, shapes, and orientations of the molecule analyte with respect to the substrate (and hot spot localization) play a key role in the assessment of the SERS reproducibility. In a recent paper, the reproducibility in parallel with the EF of three different SERS substrates has been evaluated using three representative biological probes, i.e., ultralow-molecular-weight molecules of biphenyl-4-thiol (BPT, small molecule, 186.27 Da), purified protein solutions (BSA, medium molecule, 66.5 kDa) and red blood cells (RBCs, diameter <10 μm, complex target) [88]. The considered SERS substrates showed different geometries, as summarized in Figure 4: (i) Hexagonally close-packed arrays of polystyrene nanospheres (d = 500 nm) covered with a 30 nm gold layer (CPA); (ii) Non-close arrays of gold-coated polystyrene nanospheres obtained by size reduction with the plasma of the close-packed nanospheres, named SA; (iii) Arrays of gold nanotriangles obtained by sonication of CPA, resulting in SR; [88]. The experimental tests using BPT showed high and stable SERS signals, emphasizing how the BPT molecules can arrange homogeneously on the three substrates. The numerically estimated SERS intensities expected for the three substrates confirmed that the SA and SR substrates had SERS intensities lower than those of CPA in excellent agreement with the experimental data. By analyzing the performance of the three substrates (CPA, SA, and SR) in the detection of BSA or RBCs, it has been observed that the mean SERS spectrum measured for the CPA substrate is comparable (or lower in the case of RBCs) to the spectra obtained for the SR and SA substrates and the variability much higher. Such an SNR decrease can be ascribed to a partial covering of BSA and RBCs onto the CPA substrate with respect to BPT as well as a less deterministic SERS interaction between the biological target and substrate. Indeed, BPT forms a self-assembling monolayer homogenously adsorbed onto the substrate, while the other biological targets do not assemble in an ordered fashion. Overall, the quantitative comparison revealed that the direct analysis of the performance of the SERS substrates cannot disregard the biological target of interest because the analyte size effect can systematically affect the intensity and variability of the SERS spectra [88].

Chemical specificity is an additional advantage of SERS-based sensing because the Raman peaks allow for easier distinction of different molecules and also different molecular configurations [89]. However, direct sensing, i.e., the acquisition of the SERS spectrum of the analyte molecule in close contact with the plasmonic substrate, presents another important limitation associated with the intrinsic complexity of the matrix (biological fluids such as urine, blood, saliva, cell lysates, etc.) where the target analyte is usually present at a low concentration together with a myriad of other biological components. A possible solution for measuring biological targets with good reproducibility in complex matrices relies on the use of indirect SERS sensing. In this case, detection is performed after the functionalization of the SERS substrate with reporter molecules with a unique and strong Raman fingerprint and displaying a high binding affinity and specificity with the target analyte. Thus, in indirect sensing, the concentration of the analyte is indirectly obtained by measuring the SERS spectrum of the Raman reporter engineered for selectively binding to the analyte molecule. The standard SERS detection with tag includes a core of metal nanoparticle of any shape, an efficient and photo-chemically stable Raman reporter (most of the cases dye has been used for the large cross-section), and a protective shell functionalized with targeting moieties [90,91]. Sometimes small organic molecules like benzenethiol are used as Raman reporters which strongly bind to the surface of the nanoparticles and prevent the chance of photobleaching effect as compared with dye molecules [19]. The reporter should be chosen in such a way that the signal from the reporter should not be superimposed with the Raman peak of the molecule of interest.

One interesting protocol related to indirect SERS sensing is the use of bi-orthogonal Raman reporter. In this case, the bio-orthogonality is defined as that the Raman reporter possesses a vibration in the Raman-silent region of a cell (approximately 1800 to 2800 cm^−1^), where the Raman signals from biological components are negligible [92]. Another important part is the protective shell, normally polyethyleneglycole (PEG), and silica is used as a shell. It can be functionalized with different antibodies and peptides for any specific target sensing. The protecting layer helps different objectives like preventing the nanoparticles from aggregating in the medium, maintaining the binding of the reporter, and hindering non-specific adsorption of cellular components.

The potential of these indirect methods in improving the SERS reproducibility has been explored for sensing of biological materials in complex matrices or for SERS imaging in in-vivo experiments. Even though the indirect sensing protocol is extremely useful especially for samples with small cross-section like biomolecules, it is not a label-free technique, therefore, not useful for the sensing of the unknown analyte. Along with this, a good knowledge of functionalization protocols is always essential for the fabrication of the SERS substrate during indirect sensing.

## 5. Toxicity of SERS Substrate

SERS is a powerful tool which application in biomedicine presents advantages for label-free single-cell imaging, single-cell analysis, biomarker detection, and control of drug delivery [93]. The use of SERS substrates in living cells poses issues about their safety. Generally, gold is the material of choice for SERS biomedical applications since it presents several advantages, such as easy, reproducible synthesis and optoelectronic properties that are related to AuNPs size, shape, and surface functionalization. The investigation about AuNPs toxicity produced controversial results with some works claiming AuNP safety and others demonstrating AuNPs toxicity [94,95]. Spherical AuNPs and Au nanorods are the most used in biomedical research and the most studied for their effects on eukaryotic cells. It has been reported that Au nanorods are more cytotoxic than spherical AuNPs in different kinds of human cells including human fetal osteoblast (hFOB 1.19), osteosarcoma (143B, MG63), and pancreatic duct cells (hTERT-HPNE) [96,97]. This seems to be due to the presence of cetyltrimethylammonium bromide which is essential for Au nanorod synthesis [96,98]. AuNPs size is another factor that influences cytotoxicity. As an example, Lee et al. [99] studied that smaller (5 nm) AuNPs were more cytotoxic than bigger (100 nm) AuNPs reducing the cell viability of human neuronal precursors by up to 50%. Similar results were also obtained with other cell types as human hepatocellular carcinoma (HepG2) and fetal hepatocytes (L02) cells [100]. AuNPs have also been associated with cell oxidative status imbalance in human adenocarcinoma (HT29) and HepG2 cells [101]. Anyway, AuNPs toxicity seems to also depend on the cell types; for this reason, it is highly recommended to test the toxicity of AuNPs on cells before running experiments.

Another important aspect that is worthy of note is AuNP immune safety. Indeed, AuNPs have been often reported as activator of pro-inflammatory pathways [94,95]. These pro-inflammatory effects may be due to external contaminations and not to AuNPs per se. One of the most common pro-inflammatory AuNPs contaminants is the bacterial endotoxin (lipopolysaccharide, LPS). This molecule is one of the most important components of the Gram-negative bacteria cell wall and it is ubiquitous in the environment. LPS is thermo-stable, resistant to the most used sterilization procedure and human cells are very sensitive to it. For these reasons, it is crucial to test AuNPs for endotoxin contamination before using AuNPs in experiments involving living cells [102].

## 6. Biosensing Using SERS

Label-free SERS detection, that is the measurement of the intrinsic enhanced Raman signature of the analyte molecule, or indirect SERS based-sensor has been used for the detection of different complex molecules for applications in biomedicine [103] including nucleic acids, proteins, and cells. In this section, we summarize a few cutting-edge recent reports related to SERS-based biomolecule detection using gold nanoparticles/nanostructures, as reported in the Table 2.

### 6.1. Sensing of DNA/RNA

In modern medicine, the identification of DNA and RNA has significant implications for the analysis, diagnosis, and treatment of genetic and infectious diseases. Moreover, the recent global SARS-CoV-2 pandemic, with the emergence of the COVID-19 and its multiple variants, illustrates the unmet need for rapid, flexible nucleic acid testing technology for disease diagnostics [104]. To date, RT-PCR is the gold standard oligonucleotide detection technique, which however requires several reagents and procedural steps (e.g., nucleic acid isolation and retro-transcription), expensive equipment, reagent costs, good laboratory practices, and skilled personnel [105]. Indeed, the basic RT-PCR assay relies on extraction and purification of the nucleic acid, then exponential amplification of the target sequence, using a thermostable polymerase enzyme and specific primers [105]. An important issue is also the time for the analysis, which goes from few hours to 1–2 days. The needs for designing specific primers for the target require handling by experienced operators, capable of detecting errors. The technique, if not performed in rigorous conditions and with controlled reagents, is particularly prone to false-positive results. In this matter, SERS is one of the useful promising ultrasensitive techniques to accurately identify specific DNA/RNA fragments [106,107,108,109,110,111].

Wang et al. [106] synthesized gold plasmonic nanopores by in situ reductions of gold on the confined tip of a glass nanopipette, where a bias potential was applied to drive DNA oligonucleotides and amino acids translocating through the nanopores (where the hot spots are located), which enables collection of rich structural information for biomolecules by SERS technique (see Figure 5a,b). With the application of the bias potential, the Adenine SERS signal can be monitored over a large range of concentrations from 10^−4^ to 10^−9^ M (Figure 5c). The SERS spectra of four nucleobases of A, T, G, and C display a clear fingerprint of Raman features that can be useful for the direct identification of single nucleobases in DNA oligonucleotides (Figure 5d). Indeed, the authors demonstrated that the SERS-based nanopore detection can directly provide structural information, which enables it to distinguish DNAs with a single nucleobase difference with high spatial resolution and high sensitivity.

Recently, Kim et al. [107] developed a SERS-based plasmonic biosensor for the label-free multiplex detection of miRNAs (miR-10b, miR-21, and miR-373), which are relevant to cancer metastasis. The SERS-based nanoplasmonic biosensor was fabricated with the application of a head-flocked gold nanopillar substrate and a complementary DNA probe platform was used for selective targeting of miRNAs with an extremely low detection limit, low signal fluctuations, and high signal stability. The label-free SERS nanoplasmonic biosensor was used to discriminate between target miRNAs (miR-10b, miR-21, and miR-373) by performing the SERS signal measurement according to the hybridization process with ultra-high sensitivity (LODs of 3.53 fM, 2.17 fM, and 2.16 fM, respectively). According to the report, this plasmonic biosensor offers many advantages such as label-free detection, high selectivity, high sensitivity, and multiplex detection capability.

Several indirect sensing methods have been reported in the literature for improving the detection of oligonucleotides using SERS. In most of the SERS-based DNA/RNA sensors, a Raman tag is attached with the plasmonic nanostructures, and indirect highly sensitive, and highly reproducible sensing is performed. Indeed, sandwiched DNA/RNA structures where the movement of the Raman reporter or organic dye close to or away from the plasmonic nanostructure is realized by standard hybridization leading to a double-stranded DNA (or DNA/RNA hybridization) or by the formation or decomposition of a hairpin chain (stem-loop configuration) has been demonstrated by several groups [91,108,109,110]. The use of indirect approaches produces significantly high SERS signals for the detection of DNA with a notably low concentration (<10 fM), high sensitivity, specificity, and reproducibility [108,109]. The combination of the use of high-performing SERS substrates and specific DNA targets with Raman dyes can enhance the multiplexed detection capability of the SERS sensor and crucially, providing quantitative information [110]. In a recent paper, Fabris et al. [111] demonstrated the use of the SERS approach for intracellular monitoring of RNA mutations in the influenza A virus. Gold nanostars, functionalized with a DNA hairpin structure decorated with a Raman-dye was designed to selectively extend/fold in the absence/presence of the viral RNA targets. “OFF-ON” switching of the SERS signal was observed when the fluorophore was away from or close to the gold nanostar surface (Figure 5e,f). The degree of signal recovery correlated with the number of genetic mutations has been reported. The authors for the first time employed molecular beacon-based SERS probes to detect viral RNA targets in intact individual cells. Indeed, the experiments carried out with HeLa cells transfected with plasmids coding for either hemagglutinin (HA) segment or two other segments (negative controls) demonstrate the functionality of the molecular beacon-based SERS probes in intact individual cells with high sequence sensitivity. These results suggest the applicability of these probes for multiplexed detection and quantification of viral RNAs in individual cells, including the RNA from COVID-19, with an approach that can account for the viral population diversity.

### 6.2. Sensing of Proteins

Detection and quantification of proteins in biological fluids associated with specific diseases at the point-of-care (POC) have attracted increasing interest for many diverse medical applications. State-of-the-art immunological methods require direct fluorophore labeling of the protein, labeling secondary reagents that bind to the protein, or labeling secondary reagents that bind to a tag such as biotin. These methods can be carried out with ELISA equipment making it inexpensive and readily available. However, the use of labels can reduce activity or selective binding, labeled secondary reagents are not always available, there is a limit on the number of proteins that can be detected at the same time as well as the photo-bleaching and degradation of the fluorophore. Due to these issues, there is a push in the research community to use label-free methods, for instance, SERS spectroscopy for protein detection. Obtaining SERS data from solutions of proteins is challenging as their structure is bigger than molecules, can easily denature, aggregate, modify their conformation as well as being generally present in complex matrices (biological fluids, blood, urine, cells, etc.) at low concentration [112].

One direct sensing protocol using the concept of in-liquid molecular detection via SERS has been reported by Fazio et al. In this work, the authors have implemented a label-free, all-optical SERS sensor for the detection of biomolecular in buffer solution (Phenylalanine (Phe), Bovine Serum Albumin (BSA) and Lysozyme (Lys) at concentrations down to few μg/mL) which exploits the radiation pressure to push gold nanorods on a surface and form SERS-active aggregates in buffered solutions of amino acids and proteins. The schematic of the principle of SERS detection of biomolecules using optical force has been shown in Figure 6a, also the SERS detection of BSA molecule in phosphate buffer solution has been shown in Figure 6b. Here, the estimated maximum enhancement factor was reported around 10^5^, using the excitation laser 632.8 nm. According to the authors, the main advantages of this sensor are that the detection happens in the natural environment of biomolecules, the sensing is rapid, the technique is experimentally simple, reliable, and intrinsically scalable to lab-on-chip devices [67].

Brule et al., reported the study of bovine serum albumin (BSA) protein conformations using SERS, where they used self-assembly of raspberry-like AuNPs immobilized for the fabrication of SERS substrate [113]. The substrate is excited by a 784 nm laser and the estimated maximum enhancement factor reported is around 10^7^ and the limit of detection is almost 10 pM. Also, the dynamic SERS record enables discriminating the physisorption and the chemisorption events using multivariate analysis. This event is confirmed by different characteristic SERS signals recorded from the hydrophobic amino acid fingerprints, like tryptophan, tyrosine, leucine, and histidine. According to the authors, tryptophan is a specific biomarker for the unfolded conformation of the BSA. Indeed, BSA has only two tryptophans in its structure, one is at the protein surface and the other one is located in the hydrophobic core. During BSA unfolding, the tryptophan in the hydrophobic core is exposed and is free to give a strong and specific signal, thanks to the high Raman vibrational activity of the benzoic structure. Instead, the tryptophan at the protein surface has the benzoic ring still blocked by the surrounding amino acids and unable to give vibrational signals. For these reasons, tryptophan-associated bands are considered good markers of BSA unfolding. Moreover, when BSA is unfolded also other amino acids, such as histidine, that surround the inner tryptophan are exposed and give Raman signals developing a high electrostatic affinity with gold [113].

Another indirect protein sensing platform via SERS has been reported by Zengin et al., where they present a homogeneous detection method utilizing monoclonal anti-tau functionalized hybrid magnetic nanoparticle (MNP) probes and polyclonal anti-tau immobilized AuNPs as SERS tags in solution [114]. The tau specificity on BSA and immunoglobulin G (IgG) was also tested using this technique. The schematic of the experimental procedure and the SERS spectra of monoclonal anti-tau functionalized hybrid nanoparticles exposed to BSA (500 nM), IgG (500 nM), tau (500 nM), and a solution with equal amounts (500 nM) of BSA, IgG, and tau has been shown in Figure 6c,d. According to the authors, this SERS-based sandwich assay system possesses advantages over the current methods such as ELISA, LSPR, and chromatography which contain label-free and rapid detection using a simple and cost-effective substrate fabrication technique. This approach might provide accurate, sensitive, and more selective detection of tau than current protein detection methods with the lowest limit of detection for tau solution of below 25 fM, which is comparable to the sensitivity of conventional optical biosensing methods [114].

**Figure 6 nanomaterials-11-02679-f006:**
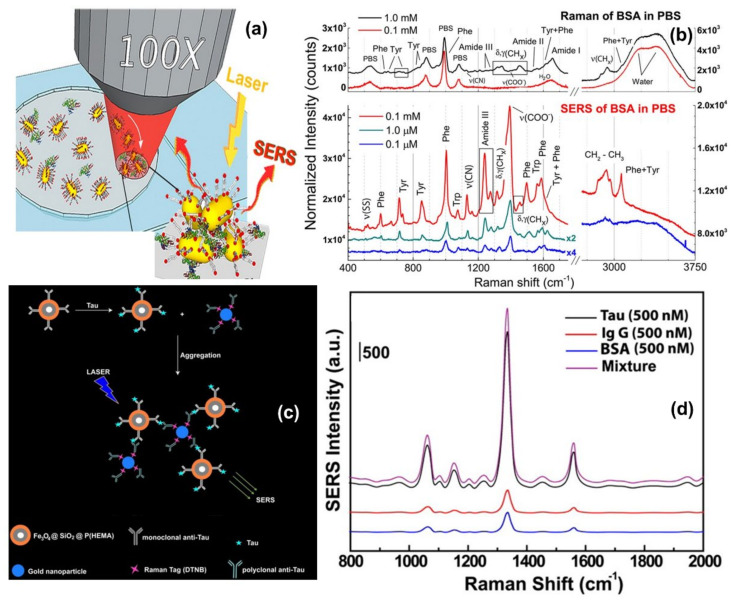
Panel (**a**) shows the schematic of the principle of SERS detection of biomolecules using optical force. Panel (**b**) shows the comparison between Raman spectra and SERS spectra of BSA and Phenylalanine in phosphate buffer solution. Reproduced with permission from Springer Nature Publications of ref. [67]. Panel (**c**,**d**) are the schematic of the experimental procedure of indirect protein sensing platform and the SERS spectra of monoclonal anti-tau functionalized hybrid nanoparticles exposed to BSA (500 nM), IgG (500 nM), tau (500 nM), and a solution with equal amounts (500 nM) of BSA, IgG, and tau. Reprinted (adapted) with permission from ref. [114], Copyright 2013 American Chemical Society.

### 6.3. Sensing of Cells

Label-free SERS-based methods can be used to detect intrinsic spectroscopic signatures from living cells that can be acquired to identify different cell components. Bando et al. [115] used 3D SERS-imaging to track AuNPs moving in the cytosol of living cells. The simultaneous detection of the SERS signal and spatial position of the nanoparticles enabled to reconstruction large amount of information on biomolecular events. Time-space-spectrum variable analysis allowed the visualization of molecular dynamics such as dissociation of proteins in biological reactions or the study of molecules associated with the transportation process such as vesicle transport, nuclear entry, and protein diffusion.

SERS fingerprinting of individual molecules enables the possibility of cancer screening. Liu et al. [116] proposed the use of economic, storable, biodegradable, and easy fabricated paper-based SERS substrate for automated, rapid, and non-invasive cancer cell screening. Different and reproducible SERS spectra were detected from normal and cancerous cells due to specific biomolecular changes in cancerous cells (shown in Figure 7a,b). A diagnostic algorithm based on band ratio analysis allowed the discrimination with a sensitivity and specificity up to 70%.

Since direct SERS-analysis and full band assignment suffer from SERS signal stability and reproducibility, SERS tags for indirect cancer detection have been additionally proposed. Wu et al. [117] developed three SERS-active AuNPs with different shapes (nanosphere, nanorods, and nanostars) modified with a Raman reporter molecule (4-mercaptobenzoic acid, 4-MBA) for detection of circulating tumor cells (CTCs) from blood. These nanoparticles showed a strong SERS signal and excellent sensitivity due to the Raman reporter (LOD of 1 cell/mL) and high specificity due to conjugation of the targeted ligand (acid folic) which can reduce the nonspecific catching or uptake by healthy cells and can recognize CTCs of various cancers overexpressing of folate receptor alpha (including ovarian, brain, kidney, breast, lung, cervical, and nasopharyngeal cancers).

Cancer cells are characterized by specific bioreceptors present in their plasma membranes. The detection and accurate identification of bioreceptor expression on the cell surface represents a crucial step in the diagnostic process. In this scenario, the high multiplexing capability of SERS has the potential to provide accurate molecular phenotyping of the individual cancer cells. Bodelón et al. [118] proposed a new class of SERS tags based on poly (N-isopropylacrylamide) (pNIPAM) encapsulated AuNPs for the multiplex detection of tumor-associated protein biomarkers. The authors demonstrated that this system overcame particle aggregation issues, allowed them to achieve high resonance Raman scattering enhancements leading to reproducible high-intensity signals, and allowed multiplex detection and imaging of three important tumor-associated biomarkers (EGFR, EpCAM, or CD44), for discrimination of A431 tumor and 3T3 2.2 nontumor cells with a single excitation laser line.

Real-time monitoring of a single cancer cell, under different conditions, and extracellular stimuli, can be assessed by coupling SERS spectroscopy with single-cell microfluidics as reported by Willner et al. [119]. A single prostate cancer cell was trapped thanks to the microfluidic device in a droplet and the developed wheat germ agglutin-functionalized SERS nanoprobes used for spectroscopic identification of glycans on the cell membrane. The analysis was performed in two steps. First, a large sample area was scanned with a low spectral resolution enabling the ‘fast’ identification of SERS regions of interest in cancer cells. In a second step, the SERS signal from WGA-modified metallic nanoparticles was used to ‘slower’ monitor the expression of glycans at a higher spectral resolution and to screen the dynamic cell-to-cell variability in a higher throughput fashion.

Intrinsic SERS signals can be used to detect specific biomarkers or physiologically relevant biomolecular species inside or near individual cells, such as important metabolites in normal and tumor cells, or other cellular components such as exosomes. The application of SERS to identify the presence of extracellular metabolites, secreted by cancer cells and relevant to tumor biology, including tryptophan, kynurenine, and purine derivatives, has been demonstrated by Plou et al. [120]. The proposed SERS-based strategy used a plasmon active substrate comprising a superlattice of Au nanoparticles, as the source of enhancement for the Raman signal from the analytes (for details see in Figure 7c–e). The authors claimed that the sensitive and cost-effective plasmonic substrate was effectively combined with 3D cell culture models, which more closely recreate the biochemical and biophysical factors in the tumor microenvironment, toward real-time imaging of heterogeneous metabolic alterations and cytotoxic effects on tumor cells, which will have significance in diagnosis and therapy.

**Figure 7 nanomaterials-11-02679-f007:**
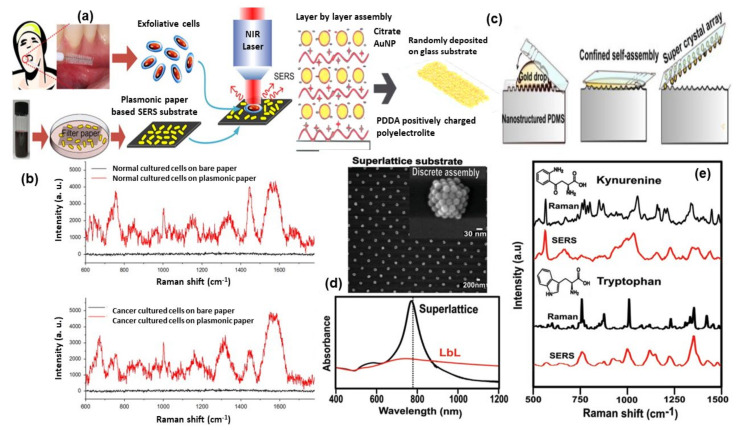
Panel (**a**) shows the illustration of paper-based SERS substrate and SERS examination of the exfoliated cells obtained from oral-cancer patients. Panel (**b**) represents the SERS spectra from the exfoliated cells: normal tissue (top) and cancerous tissue (down) [116]. Panel (**c**) shows the methodology of super-lattice SERS substrate fabrication and its SEM image and Vis-NIR spectra are shown in panel (**d**). Panel (**e**) shows the comparison of Raman and SERS spectra for kynurenine (Kyn) and tryptophan (Trp). Reproduced with permission from Wiley publications of ref. [120].

Managò et al., demonstrated one interesting way for SERS probing of red blood and leukemic cells (see Figure 8a–d). In that experiment, diatom frustules have been used which is formed by a complex, intricate but regular dielectric 3D nanostructure. The diatom frustules properly metalized trigger broadband LSPR in the red and infrared spectral range and can be used as SERS substrate for cell probing. The SERS spectra from red blood cells and leukemia cells were mainly related to the cell membranes whose impairment is responsible for several pathologies. This allowed the limitations of conventional spontaneous Raman spectroscopy, where the scattered light comes from the cell as a whole, to be overcome [121].

Quantitative and label-free detection of glycerophosphoinositol (GroPIns) using SERS techniques has been reported by De Luca et al., GroPIns are cell metabolites regulating important cell biological functions. Indeed, their increased concentration in the cell cytosol has been associated with several diseases, including thyroid cancer [122]. The detection of GroPIns cellular levels can be considered a biochemical marker of photo/physiological conditions. In their paper De Luca et al., used Au fishnet nanostructures, fabricated using e-beam lithography, as SERS active substrate for sensitive detection of GroPIns in complex matrices. The authors demonstrated that the proposed SERS substrate provided more reproducibility as compared with colloidal nanoparticles [122] together with a good enhancement. Indeed, the proposed SERS-based approach allowed for a rapid (acquisition time: 1 s), quantitative (accuracy: 6%), and a sensitive (detection limit: 200 nM) detection of the GroPIns in complex matrices eliminating the need for labels/dyes, long sample preparation and the use of large volume samples (single-cell volumes can be used).

Tian et al., recently developed a SERS sensor for the detection of exosomes, small extracellular vesicles (30–150 nm) which transfer and deliver encapsulated molecules from their originating cells, as a cancer biomarker. In the proposed indirect SERS sensor, 4-mercaptobenzoic acid (4-MBA) embedded in Au overcoated nanostars (AuNS@Au) was used as a Raman reporter, bivalent cholesterol labeled DNA anchor conjugated onto AuNS@4-MBA@Au formed SERS nanoprobes and exosome-bound magnetic beads were used as the capture probes. Exosomes are specifically captured by immune magnetic beads, and then SERS nanoprobes are fixed on the surface of exosomes by hydrophobic interactions between cholesterol and lipid membranes, thus forming a sandwich-type immune complex. The immune complex can be magnetically captured and produce enhanced SERS signals. In the absence of exosomes, the sandwich-type immune complex cannot be formed, and thus negligible SERS signals are detected. The degree of immune-complex assembly and the corresponding SERS signals are positively correlated with the exosome concentration over a wide linear range of 40 to 4 × 10^7^ particles per µL and the limit of detection is as low as 27 particles per µL [123].

SERS-based sensors can be used to monitor certain physiological processes as well as the real-time release of drugs in living cells [124]. Managò et al., recently demonstrated the use of hybrid nanoparticles, consisting of diatomite nanoparticles decorated with gold nanospheres and covered by a pH-sensitive gelatin shell, for label-free imaging of the nanovector localization and local SERS-sensing of the intracellular release of galunisertib in living colorectal cancer cells (see Figure 8e–g). AuNPs strongly enhance the SERS signal of the drug-loaded in the diatomite pores allowing the tracing and quantification of its release in living cells over days with high sensitivity (down to sub-femtogram of the drug). Furthermore, the encapsulation of galunisertib in the nanoplatform can help to lower the amount of drug required to inhibit the metastatic process in cancer cells, reducing the formation of drug-related toxic metabolites.

**Figure 8 nanomaterials-11-02679-f008:**
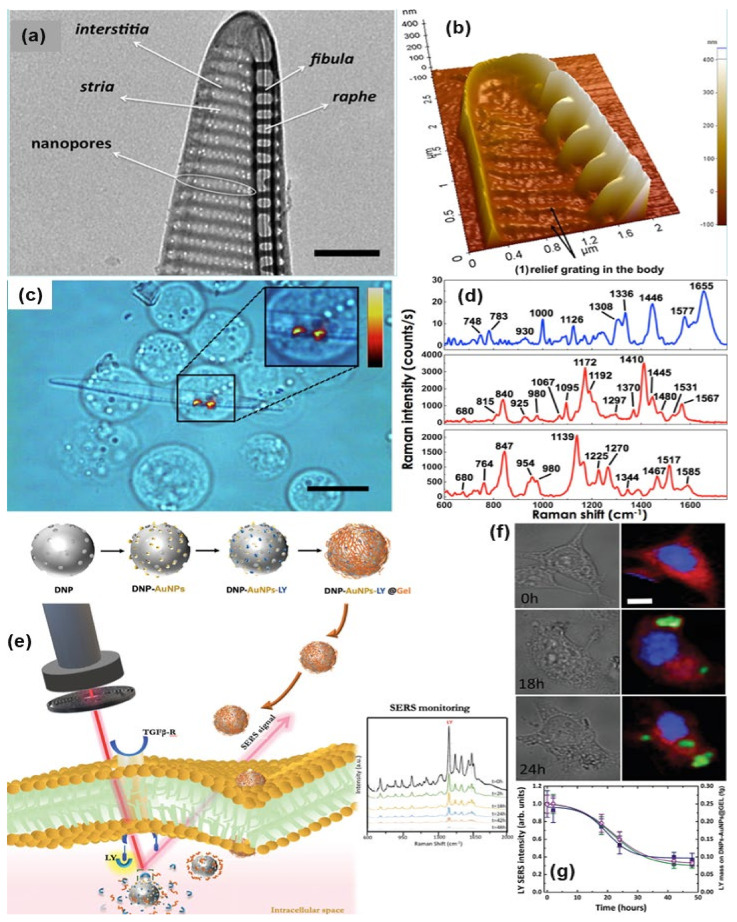
TEM and AFM image of a P. multistriata single valve is shown in panels (**a**,**b**). Panel (**c**,**d**) shows the optical image of REH LCs over a single metalized diatom valve and the conventional Raman spectrum of a single REH cell. Reprinted (adapted) with permission from ref. [121], Copyright 2018 American Chemical Society. The schematic diagram of hybrid nanosystem (diatomite nanoparticles-AuNPs-LY@Gel) synthesis and internalization in colorectal cancer (CRC) cells have been shown in panel (**e**). Panel (**f**,**g**) show optical image and Raman mapping images showing the internalization of DNP-AuNPs-LY@Gel (50 μg mL^−1^) into CRC cells after 0, 18, and 24 h of incubation (scale bar = 10 μm) and time-dependent LY SERS signal from the hybrid nanocomplex in living CRC cells. Reproduced with permission from Wiley publications of ref. [124].

**Table 2 nanomaterials-11-02679-t002:** A brief summary on biosensing using SERS.

Type of Nanoparticles/ Substrate	Target	Type	Detection Mechanism	Enhancement Factor (EF) or Limit of Detection (LOD)	Ref.
Au nanopores	DNA	Direct	Bias potential + SERS detection of DNA	EF: 10^6^	[106]
Au nanopillar	miRNA	Direct	DNA/RNA hybridization + SERS detection of DNA/RNA	LOD: 3.53 fM	[107]
Au NPs-decorated silicon nanowire array	DNA	Indirect	stem-loop DNA/target DNA hybridization + SERS detection of dye molecule	EF: 7.24 × 10^5^ LOD: 10 fM	[108]
Spherical Au nanoparticles	miRNA	Indirect	Symmetric signal amplification + SERS detection of Cy3	LOD: 7.5 fM	[110]
Au nanostar	RNA	Indirect	Identify and quantify RNA mutations through SERS	/	[111]
Au nanorod	BSA	Direct	Optical Tweezers + SERS detection of BSA	EF: 10^5^	[67]
Raspberry-like assembled Au nanoparticles	BSA	Direct	Dynamic SERS of BSA	EF: 10^4^–10^7^ LOD: 10 pM	[113]
Au NPs + magnetic NPs	tau protein	Indirect	SERS-based sandwich assay	LOD: 25 fM	[114]
Au NPs	Hela cells	Direct	Intracellular detection of proteins/cytosol by SERS	/	[115]
Au nanorods	Oral cancer cell	Direct	Cancer cell screening using SERS	Sensitivity: 70% Specificity: 60%	[116]
Au nanosphere, rods, stars	Circulating tumor cells	Indirect	SERS detection of Raman reporter (4-MBA) for identification of cancer cells	EF: 10^4^	[117]
Au octahedral NPs	Tumor Cells	Indirect	Detection and imaging of cancer cells by SERS tags	/	[118]
Functionalized Au NPs	Prostate cancer cell	Indirect	Imaging and identification of Glycans in cell membrane by detection of the SERS probe	/	[119]
Au superlattices	tumor metabolites	Direct	Identification of cell metabolites by SERS on a chip device	/	[120]
Au metallized diatom	red blood, leukemic cells	Direct	SERS detection of cell membrane	EF: 10^6^	[121]
Au fishnet	Cell metabolites	Direct	SERS detection of Glicerophosphoinositol	LOD: 200 nM	[122]
Au nanostars	Exosomes	Indirect	SERS-based sandwich assay	LOD: 4 × 10^4^ particles per µL	[123]
Diatomite nanoparticles decorated with Au NPs	Drugs in colorectal cancer cells	Direct	SERS-detection of Drugs in living cells	EF: 10^5^	[124]

## 7. Conclusions

In this review, we have described the use of Au nanostructures-based SERS substrates for applications in the biosensing domain. The main characteristics of a SERS-based biosensor are high chemical specificity and strong sensitivity. Indeed, it has been used for detection even for very low concentrations of various biomolecules, including DNA, microRNA, protein, bacteria, cells, and whole blood, providing important structural information, as summarized in Table 2. The key element of the sensitivity of the SERS signal is the selection of the SERS substrate and the formation of the hotspot which directly influences the cross-section of the probe molecule. This hotspot engineering is possible by controlling the generation of LSPR, or by tuning the size and shape of the nanoparticles. To improve the SERS sensing performances, there are many metal nanoparticles and 2d materials which have the potential as SERS active substrate, however, for most of the in vivo biological SERS experiments, Au is used due to its nontoxic and chemically stable nature. From the biosensing perspective, the SERS substrate should be chemically stable, sensitive, and reproducible, along with rapid and accurate detection, good selectivity, small sample volume, low cross-contamination, and simple operation, which should be highly prioritized. Even though the plasmon active substrate plays a major role in SERS biosensing, however fabricating a SERS substrate with good reproducibility is the main limitation in case of commercialization which requires the mass production of the substrate and on a large area. Substrates produced with lithography (not large area) are reproducible, whereas the colloidal nanoparticles can produce higher enhancement factors and the analyte size effect can be controlled. Therefore, very specific methods are required to resolve these challenges for the production of standard as well as reliable SERS biosensors with ultra-sensitivity and reproducibility. To date, researchers have explored different strategies for improving the SERS-based biosensing reproducibility, including the use of indirect sensing protocols. These protocols help for quantification and identification of multiple molecules/components within complex mixtures solution, and these are the crucial steps towards biomedical applications. Indeed, SERS has been used in the field of bioimaging, disease diagnosis, drug delivery, and single-cell detection and identification. These studies are highly promising and demonstrate the strong capability of the SERS approach for biosensing.

Another important issue of the SERS-based sensing technique towards clinical diagnostic technology from medical research is the requirement of sophisticated instruments. Part of this problem can be resolved by the use of microfluidic channels attached with SERS tags and a portable Raman device. These pave the way for automated analysis. The low-cost and miniaturized SERS-based biosensing smart device which is capable of point of care testing and easy integration with other devices will bring a new dimension in future research.

## Figures and Tables

**Figure 1 nanomaterials-11-02679-f001:**
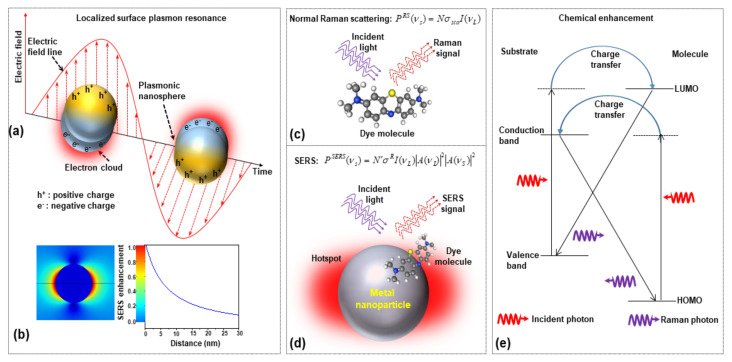
The schematics of (**a**) localized surface plasmon resonances (LSPR) from plasmonic nanoparticles, (**b**) numerical simulation of the electric field distribution of isolated AuNP and its dependence of SERS enhancement on the distance from nanoparticle surfaces, (**c**) the normal Raman scatting process, (**d**) two-step EM enhancement mechanism in SERS, and (**e**) chemical enhancement in SERS (HOMO: highest occupied molecular orbital and LUMO: lowest unoccupied molecular orbital).

**Figure 2 nanomaterials-11-02679-f002:**
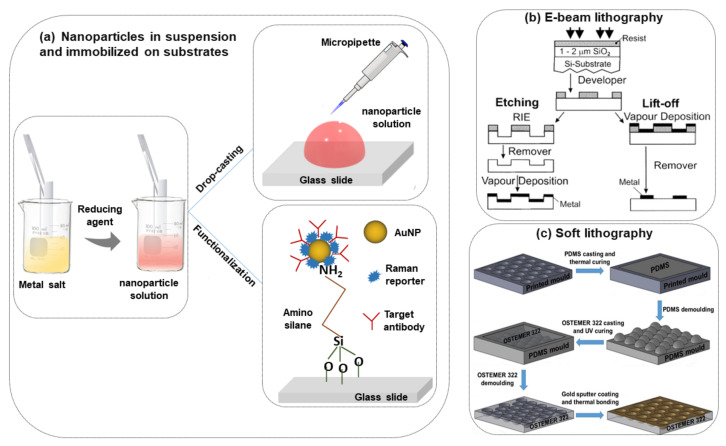
Scheme of the different procedures of SERS substrate fabrication, (**a**) nanoparticles in suspension and immobilized on the solid substrate [6], (**b**) e-beam lithography [32], (**c**) soft lithography [33].

**Figure 3 nanomaterials-11-02679-f003:**
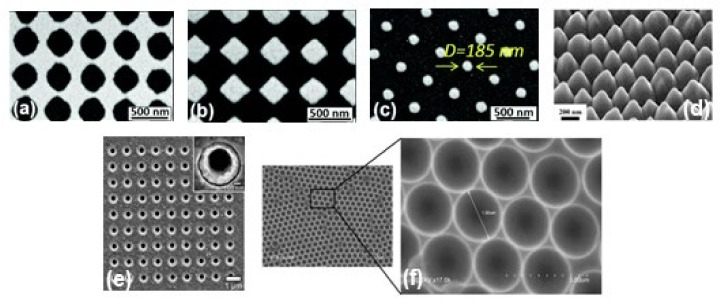
SEM images of metallic periodic nanostructures nanohole arrays (**a**), diamond-shaped nanoparticle arrays (**b**), arrays of cylindrical nanoparticles (**c**) fabricated by different development times 95 s, 110 s, and 165 s, respectively. Reproduced from ref. [73] with permission from the Royal Society of Chemistry. (**d**) 45° tilt view of nanostructure arrays with Au coating thickness 60 nm. Reproduced with permission from Springer Nature Publications of ref. [49]. (**e**) nanohole array with 500 nm diameter, the inset shows the magnified image of one of the holes. Reprinted (adapted) with permission from ref. [70], Copyright 2010 American Chemical Society. (**f**) the cavity array substrate of 2 μm fabricated applying polystyrene microsphere template approach [33].

**Figure 4 nanomaterials-11-02679-f004:**
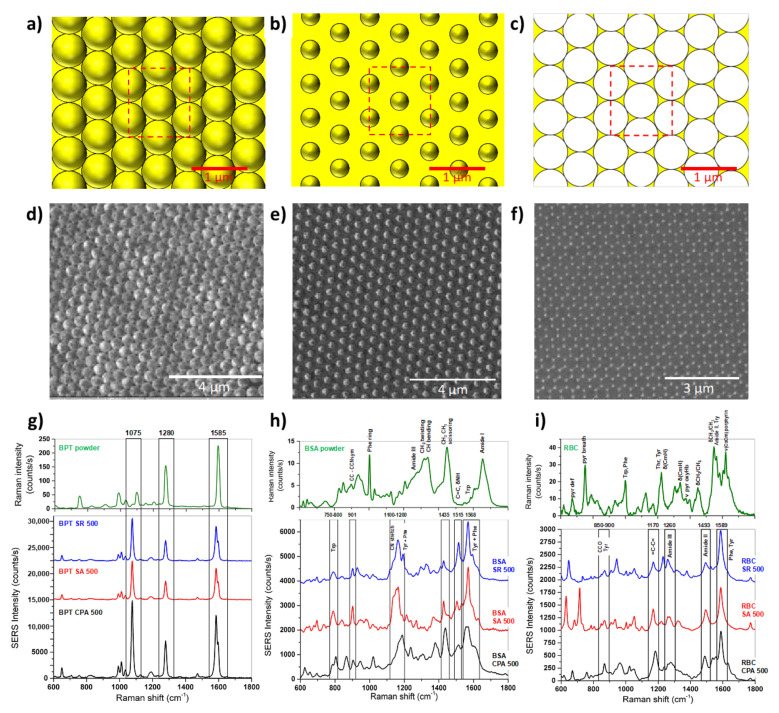
Schematic of the nanosphere periodic hexagonal patterns of CPA (**a**), SA (**b**), and SR (**c**). [Dimension: CPA (*p* = 500 nm, d = 500 nm, tAu = 30 nm), SA, (*p* = 500 nm, d = 350 nm, tAu = 30 nm), and SR, (tAu = 30 nm). Focused ion beam micrographs of CPA (**d**), SA (**e**), and SR (**f**) gold structures featuring a 500 nm period and hexagonal tile. Raman spectrum (green) of BPT (**g**), BSA (**h**), and RBC (**i**) and mean SERS spectra of BPT (**g**), BSA (**h**), and RBC (**i**) on the CPA (black), SA (red), and SR (blue) substrates [88].

**Figure 5 nanomaterials-11-02679-f005:**
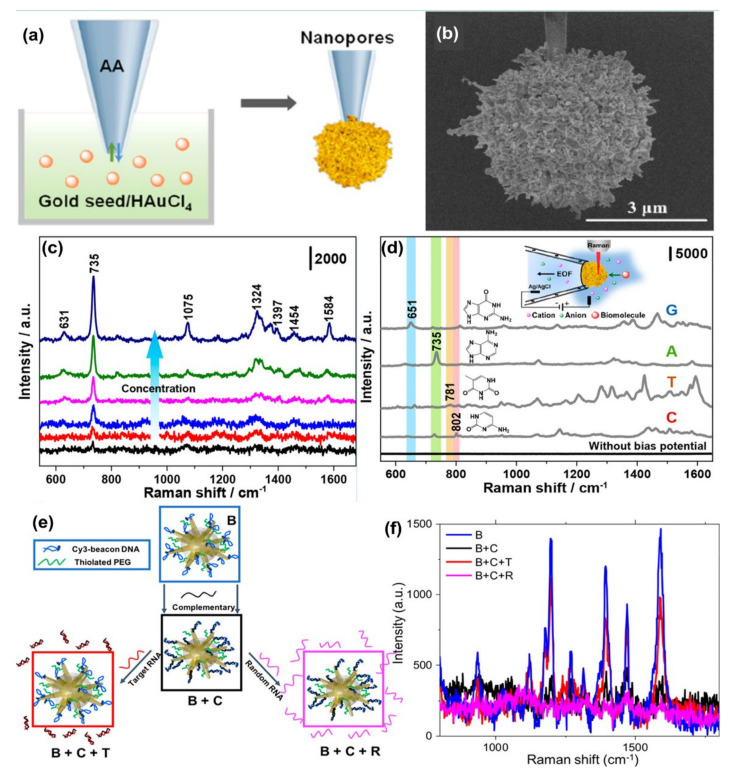
Panel (**a**) is the schematic of the synthesis of plasmonic nanopores process. SEM image of the gold plasmonic nanopores is shown in panel (**b**). Panel (**c**) represents SERS spectra of A translocating through gold plasmonic nanopores with a bias potential of −1 V, from low to high 10^−9^ to 10^−4^ M, respectively. Panel (**d**) represents the SERS-based nonresonant molecules detection of the four nucleobases (G, A, T, and C). The scheme of measurement setup (inset). Reprinted (adapted) with permission from ref. [106], Copyright 2019 American Chemical Society. Panel (**e**) shows the Au nanostars are functionalized with a Cy3-tagged beacon DNA, creating the SERS-active nanostar probes (B), and then hybridized with a complementary oligonucleotide (B + C). Upon exposure to the viral RNA targets, the complementary oligonucleotide dehybridizes from the beacon and hybridizes with the viral RNA (B + C + T), returning the beacon to its original hairpin conformation and leading to SERS signal recovery (B + C + R), Panel (**f**) is the SERS signal “ON-OFF-ON” switching: ON with beacon in hairpin conformation (B); OFF when the beacon is hybridized with 500 nM complementary oligo (B + C); then ON again upon exposure of SANSPs to 500 nM target viral RNA (B + C + T); Signal recovery was not observed after a random RNA sequence (500 nM) was introduced (B + C + R). Reprinted (adapted) with permission from ref. [111], Copyright 2020 American Chemical Society.

**Table 1 nanomaterials-11-02679-t001:** Different shapes colloidal SERS active AuNPs.

Shape	Dimension	Detected Molecule	Laser	EF	Ref.
Nanoflower	400 nm	Rhodamine 6G	785 nm	10^5^	[43]
	45 nm	Rhodamine 6G	532 nm	10^6^	[44]
Nanostar	105 nm	Crystal violet	785 nm	10^7^	[45]
	130 nm	4-mercaptobenzoic acid (MBA)	785 nm	10^9^	[46]
Nano-bowtie	Height: 40 nm, Gap: 8 nm, Edge: 90 nm	Trinitrotoluene (TNT)	785 nm	10^5^	[47]
	Height: 25 nm, Gap: 8 nm, Edge: 100 nm	Bi-(4-pyridyl) ethylene (BPE)	785 nm	10^7^	[48]
Nanorod	Length: 69 nm, Width: 24 nm	Rhodamine 6G	532 nm	10^6^	[49]
	Length: 41 nm, Width: 18 nm	Rhodamine 6G	633 nm	10^7^	[50]
Nanocube	Edge length: 170 nm	Rhodamine 6G	633 nm	10^5^	[51]
	32–72 nm	Crystal violet	785 nm	10^6^	[52]
	Edge length: 84 nm diameter 55 nm	1,8-octanedithiol (C8DT)	785 nm	10^10^	[53]
Nanosphere	Size range: 15–40 nm	Crystal violet	633 nm	10^3^–10^4^	[54]
	36.5 ± 6.3 nm	4 ATP	785 nm	10^5^	[55]
Nanospheroid	semi-major axis: 55 nm, semi-minor axis: 30 nm	Bi-(4-pyridyl) ethylene (BPE)	633 nm	10^5^	[56]
	50 nm	Crystal violet	514 nm	10^8^	[57]
Nanoshell	Core: 46 nm, Shell: 22 nm	Thiobenzoic acid (TBA)	785 nm	10^5^	[58]
Hollow nanoshell	Diameter 250 nm	Rhodamine 6G	785 nm	10^5^	[59]
	Diameter 233 ± 14 nm	Methylene blue	532 nm	10^5^	[60]

## Data Availability

The data presented in this study are available on request from the corresponding author.

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
