# Peer review of "Biosensing Using SERS Active Gold Nanostructures"

_nanomaterials, 2021, doi:10.3390/nano11102679_

Round 1
Reviewer 1 Report
Authors and co-worker's review entitled "Biosensing using SERS active gold nanostructures." providing an overview of the recent significant aspect in the field of surface-enhanced Raman Spectroscopy(SERS). For carefully reading, I recommend this review is suitable to publish in this journal. The following are some editing problems that maybe improve the work.
Editing comment:
(1) Figure 1 and 6 do not seem clear enough; the words inside the picture have lost their resolution.
(2) In Figure 3, the sub-title in "a" to "c" shows the full-brackets, but d to "i" indicates the semi-brackets. Please unify the format.
(3) In Figure 6, there is numerous information in this Figure, and it looks very complex and makes the readers feel confused. I suggest the authors may consider dividing the picture into several pieces for display.
(4) In conclusion, the authors indicate the advantages of Au nanostructures-based SERS substrate and provide numerous applications in biosensing. Except for the point of care testing, the authors could also list the possible future progress and perspective or which drawbacks, if having, should be conquered.
(5)In line 884, what is the meaning of "lost"-cost and miniaturized SERS-based biosensing smart device. Do you mean "low"-cost?
Author Response
Reviewer 1:
***********************************************
Authors and co-worker's review entitled "Biosensing using SERS active gold nanostructures." providing an overview of the recent significant aspect in the field of surface-enhanced Raman spectroscopy (SERS). For carefully reading, I recommend this review is suitable to publish in this journal. The following are some editing problems that maybe improve the work.
Editing comment: (1) Figure 1 and 6 do not seem clear enough; the words inside the picture have lost their resolution.
Response: The quality of figure 1 and 6 are modified in the revised manuscript.
(2) In Figure 3, the sub-title in "a" to "c" shows the full-brackets, but d to "i" indicates the semi-brackets. Please unify the format.
Response: It is modified in the revised one.
(3) In Figure 6, there is numerous information in this Figure, and it looks very complex and makes the readers feel confused. I suggest the authors may consider dividing the picture into several pieces for display.
Response: We have split Figure 6 to avoid confusion.
(4) In conclusion, the authors indicate the advantages of Au nanostructures-based SERS substrate and provide numerous applications in biosensing. Except for the point of care testing, the authors could also list the possible future progress and perspective or which drawbacks, if having, should be conquered.
Response: Conclusion is modified as per the reviewer’s comment.
(5) In line 884, what is the meaning of "lost"-cost and miniaturized SERS-based biosensing smart device. Do you mean "low"-cost?
Response: We are thankful to the reviewer for finding the spelling mistake, the proper word is incorporated in the modified version.

Reviewer 2 Report
The submitted manuscript reviews work in the field of SERS and biosensing. The manuscript contains useful information however the organization and overall emphasis needs to improved before it can be considered for publication:
- My overall comment is that the title and subject of this review is "Biosensing using SERS..." - it is NOT a review about SERS itself (of which there are many, some of which were referenced). The emphasis should be on the biosensing aspects and not standard Raman/SERS theory.
2. Section 2.1 (incorrectly labeled 3.1) on Raman/SERS is too verbose and must be condensed. The equations are well-covered in other reviews (some recent) and by now are even textbook material that should not be fully repeated. In addition, Equation 5 seems to be missing a geometric factor.
3. Section 3 is once again too verbose and must be condensed. Nanofabrication and its many variations is not the point of this review and is already extensively covered elsewhere in articles and even books. Instead of copious text, it would be very helpful to the reader if simple schematic diagrams were provided of fabrication processes discussed such as the colloidal lithography/mask approach shown in both Figs. 2 and 3. This would help make the figures showing SERS substrates much clearer and understandable to the general and expert reader alike.
4. Section 4 is redundant and should be incorporated into the other sections
5. Figure 6 is much too large to be useful. It is advisable to break this up into separate figures and discuss them in turn as required. Right now it is too difficult to follow. Such a figure is better suited near the introduction as a sort of "graphical abstract" or overview of the review - this will help improve the overall flow of the manuscript for the reader.
6. In my opinion, Section 7 should be the emphasis of the review. In addition, the earlier sections should also be explicitly tied to biosensing. For example, instead of "7. Applications of SERS" use a title that is specific to biosensing. Similar comments apply to sections 3, 5 and 6.
Author Response
Reviewer 2:
***********************************************
The submitted manuscript reviews work in the field of SERS and biosensing. The manuscript contains useful information however the organization and overall emphasis needs to improve before it can be considered for publication:
- My overall comment is that the title and subject of this review are "Biosensing using SERS..." - it is NOT a review about SERS itself (of which there are many, some of which were referenced). The emphasis should be on the biosensing aspects and not standard Raman/SERS theory.
Response: We modified the manuscript as per the comment. The paragraph concerning the “technique” has been strongly reduced and section 4 is now focused on the performances of SERS-based biosensors.
- Section 2.1 (incorrectly labeled 3.1) on Raman/SERS is too verbose and must be condensed. The equations are well-covered in other reviews (some recent) and by now are even textbook material that should not be fully repeated. In addition, Equation 5 seems to be missing a geometric factor.
Response: Section 2 is revised and condensed. The geometrical factor is added in the revised manuscript (see eq. 2 page 3)
- Section 3 is once again too verbose and must be condensed. Nanofabrication and its many variations are not the point of this review and are already extensively covered elsewhere in articles and even books. Instead of copious text, it would be very helpful to the reader if simple schematic diagrams were provided of fabrication processes discussed such as the colloidal lithography/mask approach shown in both Figs. 2 and 3. This would help make the figures showing SERS substrates much clearer and understandable to the general and expert reader alike.
Response: Section 3 has been condensed and a new figure (Figure 2) added providing information on the three main fabrication processes discussed.
- Section 4 is redundant and should be incorporated into the other sections
Response: Section 4 and 5 are merged.
- Figure 6 is much too large to be useful. It is advisable to break this up into separate figures and discuss them in turn as required. Right now it is too difficult to follow. Such a figure is better suited near the introduction as a sort of "graphical abstract" or overview of the review - this will help improve the overall flow of the manuscript for the reader.
Response: We have split Figure 6 to avoid the difficulties.
- In my opinion, Section 7 should be the emphasis of the review. In addition, the earlier sections should also be explicitly tied to biosensing. For example, instead of "7. Applications of SERS" use a title that is specific to biosensing. Similar comments apply to sections 3, 5, and 6.
Response: Modification has been done as per the comment.

Round 2
Reviewer 2 Report
The author's have put considerable effort into improving the manuscript. It reads better now and and is well-organized. It should be published.